# Efficient degradation of various emerging pollutants by wild type and evolved fungal DyP4 peroxidases

Khawlah Athamneh[1], Aysha Alneyadi[2], Aya Alsadik[1], Tuck Seng Wong[3,4], Syed Salman Ashraf[1,5]*

**1** Department of Biology, College of Arts and Sciences, Khalifa University, Abu Dhabi, United Arab Emirates, **2** Department of Biology, College of Sciences, UAE University, Al Ain, United Arab Emirates, **3** Department of Chemical & Biological Engineering and Advanced Biomanufacturing Centre, University of Sheffield, Sir Robert Hadfield Building, Sheffield, United Kingdom, **4** National Center for Genetic Engineering and Biotechnology, Khlong Luang, Pathum Thani, Thailand, **5** Center for Biotechnology (BTC), Khalifa University of Science and Technology, Abu Dhabi, United Arab Emirates

* syed.ashraf@ku.ac.ae

**Data Availability Statement:** All relevant data are within the paper and its Supporting Information files.

## Abstract

The accumulation of emerging pollutants in the environment remains a major concern as evidenced by the increasing number of reports citing their potential risk on environment and health. Hence, removal strategies of such pollutants remain an active area of investigation. One way through which emerging pollutants can be eliminated from the environment is by enzyme-mediated bioremediation. Enzyme-based degradation can be further enhanced via advanced protein engineering approaches. In the present study a sensitive and robust bioanalytical liquid chromatography-tandem mass spectrometry (LCMSMS)-based approach was used to investigate the ability of a fungal dye decolorizing peroxidase 4 (DyP4) and two of its evolved variants—that were previously shown to be $H_2O_2$ tolerant—to degrade a panel of 15 different emerging pollutants. Additionally, the role of a redox mediator was examined in these enzymatic degradation reactions. Our results show that three emerging pollutants (2-mercaptobenzothiazole (MBT), paracetamol, and furosemide) were efficiently degraded by DyP4. Addition of the redox mediator had a synergistic effect as it enabled complete degradation of three more emerging pollutants (methyl paraben, sulfamethoxazole and salicylic acid) and dramatically reduced the time needed for the complete degradation of MBT, paracetamol, and furosemide. Further investigation was carried out using pure MBT to study its degradation by DyP4. Five potential transformation products were generated during the enzymatic degradation of MBT, which were previously reported to be produced during different bioremediation approaches. The current study provides the first instance of the application of fungal DyP4 peroxidases in bioremediation of emerging pollutants.

## Introduction

Our modern lifestyle is intricately linked to the production and use of various chemical substances in different industrial sectors. Unfortunately, most of these substances will end up in

**Funding:** Generous support (CIRA-2020-046 and FSU-2019-09) to SSA from Khalifa University is graciously acknowledged. We also thank EPSRC (EP/E036252/1), the Open Project Funding of the State Key Laboratory of Bioreactor Engineering (to TSW), the Royal Academy of Engineering (the Leverhulme Trust Senior Research Fellowship; to TSW; LTSRF1819\15\21) and the NSTDA Visiting Professorship (to TSW) for financial support. The funders had no role in study design, data collection and analysis, decision to publish, or preparation of the manuscript.

**Competing interests:** The authors have declared that no competing interests exist.

the environment, which will ultimately lead to increased ecological pollution. Such pollution has a significant impact on human health and ecosystems specially when such chemicals are improperly disposed into water bodies [1]. These chemicals of emerging concern are called emerging pollutants (EPs) and are primarily made up of pharmaceuticals, pesticides, personal care products, dyes, and industrial chemical wastes that are found in the environment at a very low concentrations, but have the potential to cause severe effects on human and other living organisms [2, 3]. An increasing number of studies have reported on the disturbing presence of various EPs including pharmaceuticals in different water bodies [4, 5]. Pharmaceuticals and antibiotics in particular are of a serious concern due to the potential consequence of causing antimicrobial resistance [2, 6].

One example of manufacturing chemicals that is widely used as vulcanization accelerator in rubber industry is 2-mercaptobenzothiazole (MBT) [7, 8]. MBT is detected in surface water and tannery wastewater of rubber additive manufacturers [9, 10]. The accumulation of MBT in the environment is a major concern due to its toxicity against microorganisms [11] and humans [12], as well as potential carcinogenicity [9, 13]. Detection and removal strategies of such pollutants remain an area of continuous investigations. Significant advancement has been achieved in developing various remediation technologies to efficiently remove EPs from water. These technologies include approaches such as adsorption, advanced oxidation processes, hydrolysis processes and phytoremediation [14, 15]. Biological technologies based on biofilm-based reactors and activated sludge have also been reported and have gained attraction for EPs remediation due to their potential advantages such as cost effectiveness and environmental friendliness [16, 17].

Enzymatic-mediated degradation is another biological technology that is widely being developed to degrade EPs by exploiting oxidative and hydrolyzing enzymes isolated from eukaryotes and microorganisms. This *in vitro* enzymatic approach is an attractive option as it allows for a "less complex" bioremediation system where mechanistic aspects of such degradation processes can be studied and controlled carefully [18]. Laccases and peroxidases are among the most commonly employed enzymes that have are being explored to degrade various classes of EPs [19, 20].

Dye decolorizing peroxidases (DyPs) comprise a novel class of heme-containing peroxidases, which is not related to animal or plant classes of peroxidases. However, like other heme-peroxidases, they utilize hydrogen peroxide ($H_2O_2$) to catalyze the oxidation reaction. DyPs make-up a rapidly growing family of peroxidases that have so far been identified in fungi, bacteria and archaea, and exhibit both oxidative and hydrolytic activities [21–23]. They were first discovered in fungi, and they were named after their ability to degrade a wide range of dyes [24], however, the physiological role of DyPs is thought to be related to their activity towards lignin degradation [25, 26]. As such, DyPs have gained attention as potential candidates for biotechnological applications including bioremediation.

A promising DyP that is gaining a lot of attention for potential bioremediation applications is DyP4 (from *Pleurotus ostreatus* strain PC15, oyster mushroom), that was the first reported fungal DyP with the ability to oxidize manganese (II) [27]. However, inactivation of peroxidases by their co-substrate $H_2O_2$ is known be a major hurdle to commercial and industrial exploitation of these versatile enzymes, as high concentrations of $H_2O_2$ can irreversibly oxidize key amino acid residues in their active sites [28]. As with other DyPs, DyP4 is also inhibited by high concentrations of $H_2O_2$ [29, 30], however, it may be possible to enhance $H_2O_2$ tolerance of DyP4 using enzyme-engineering strategies. For example, Alessa et al. have used directed evolution and the bacterial extracellular protein secretion system to evolve DyP4. After iterative rounds of random mutagenesis, two evolved variants of DyP4 (DyP4-3F6 and DyP4-4D4)

were obtained, which exhibited significantly higher $H_2O_2$ tolerance compared to wild type (WT) [31].

In the current study, we report for the first time the ability of DyP4-WT and its evolved variants, 3F6 and 4D4, to degrade a panel of 15 diverse EPs, belonging to different chemical and pharmaceutical classes.

## Material and methods

### Chemicals

All the EPs used in the current study, 2,2'-azino-bis (3-ethylbenzothiazoline-6-sulfonic acid) diammonium salt (ABTS), hydroxybenzotriazole, (HOBT), isopropyl β-d-1-thiogalactopyranoside (IPTG), nutrient broth, glycerol, and CelLytic B Cell Lysis Reagent were purchased from Sigma-Aldrich (St. Louis, MO, USA). LCMS-grade solvents including water and acetonitrile, and hydrogen peroxide (30% w/v) were purchased from Millipore (Burlington, MA, USA). LCMS-grade formic acid was purchased from Fisher Chemical (Hampton, NH, USA), while kanamycin was purchased from Abcam (Cambridge, UK).

### LCMSMS method development

LCMSMS (SCIEX Triple Quad™ 3500, Framingham, MA, USA) was used for EP quantification in the degradation experiments. For that, a sensitive and selective method was used in the multiple reaction monitoring (MRM) mode to simultaneously detect and quantify 15 EPs in a mixture. Details of the MRM-based LCMSMS method, which specifically monitors the precursor to product transitions for each compound for quantitative analysis, are described previously [32].

Briefly, each EP was first manually tuned using direct syringe infusion pump to identify the precursor peak and the best possible products peak upon applying elevating collision energy's (CE) and declustering potential (DP) volts to ensure appropriate fragmentation of the ions without clustering for better detection. After finding the best "precursor-to-product" ion transitions, all the EPs were mixed and were ran at 0.25 ppm concentration through the liquid chromatography C18 Kinetex column (Phenomenex, Torrance, CA, USA) maintained at 40°C (2.6 μm, 100 A, 100 x 2.1 mm), to optimize the separation of the EPs. The MRM parameters of the 15 EPs are shown in Table 1. A dual-polarity electrospray ionization (ESI) source was used to ionize the eluted compounds in both positive and negative polarity modes. The mass spectrometry (MS) operating parameters were as follows: ion spray voltage 5500V, ion source gas 50psi, and source temperature 350°C.

### DyP4 expression and extraction

The plasmids carrying the genes encoding DyP4 WT (GenBank: KP973936.1) and its evolved variants, 3F6 and 4D4, were transformed into *E. coli* BL21(DE3), as previously reported [31]. The transformants were cultivated in Nutrient Broth containing 50 μg/mL kanamycin. When the $OD_{600}$ of the culture media reached 0.6, protein expression was induced by adding 100 μM IPTG and the culture was left to grow overnight at 25°C with shaking. The mutants used in the study were created using error prone PCR, as previously described [31]. The 3F6 variant was based on WT-DyP4, but had three amino acid substitutions (K109R, N312S, I56V) and four silent mutations D241 (T→C), I444 (C→T), G73 (T→A), L245 (G→T). The 4D4 variant had four amino acid substitutions—I56V, K109R, N227S and N312S, along with the same four silent mutations: D241 (T→C), I444 (C→T), G73 (T→A), L245 (G→T).

**Table 1. Summary of the name, molecular structure and LCMSMS parameters for the EPs used in the current study.**

| Emerging pollutant | RT (min) | Q1 > Q3 (m/z) | CE (V) | DP (V) | Polarity |
|---|---|---|---|---|---|
| 2-Mercaptobenzothiazole | 10.0 | 168 > 135 | 20 | 80 | Positive |
| Paracetamol | 3.3 | 152 > 110 | 22 | 80 | Positive |
| Furosemide | 10.4 | 329 > 285 | -22 | -90 | Negative |
| Methylparaben | 9.0 | 153 > 121 | 20 | 80 | Positive |
| Sulfamethoxazole | 8.7 | 254 > 156 | 21 | 80 | Positive |
| Salicylic acid | 8.8 | 137 > 93 | -22 | -90 | Negative |
| Ampicillin | 7.4 | 350 > 106 | 22 | 80 | Positive |
| Penicillin | 7.9 | 335 > 160 | 30 | 80 | Positive |
| caffeine | 7.3 | 195 > 138 | 27 | 80 | Positive |
| Chloramphenicol | 9.2 | 321 > 152 | -22 | -90 | Negative |
| Ibuprofen | 13.2 | 205 > 161 | -10 | -65 | Negative |
| 4-chloro-2-methyl-phenoxyacetic acid | 11.2 | 199 > 141 | -20 | -90 | Negative |
| Trimethoprim | 7.5 | 291 > 230 | 32 | 80 | Positive |
| Hydrochlorothiazide | 4.6 | 296 > 269 | -27 | -90 | Negative |
| Phenytoin | 10.4 | 253 > 182 | 25 | 80 | Positive |

RT: Retention time, Q1>Q3: Precursor to products masses, CE: Collision Energy, DP: Declustering Potential.

Bacterial cells were lysed using CelLytic B Cell lysis reagent according to the manufacturer's protocol. Briefly, cells were spun down and the pellet was resuspended in the lysis buffer. After 5 minutes, each lysate was spun, and the supernatant was collected in new tube which was used for further analysis. To check their activity, ABTS oxidation test was done for the induced lysates as well as the un-induced ones (control). For degradation studies, typically 70 μg (20 μL) of lysate was used for 1 mL reactions, which corresponded to approximately 21 U/mL of peroxidase (ABTS oxidizing activity). To avoid issues with differential expression levels of the three DyP4 forms, equal amounts of enzyme activity (for all the three variants) were used in all the degradation experiments.

## pH and temperature optimization

Universal buffers with pH values ranging from 2 to 9 were prepared using 0.1 M citric acid and 0.2 M sodium phosphate dibasic ($K_2HPO_4$). The activity of WT, 3F6 and 4D4 (70 μg) with different pH buffers was assessed based on the oxidation of ABTS [1 mM] to identify the optimum pH for each of the DyP4 tested. $H_2O_2$ [0.25 mM] was used to initiate the reaction and the kinetics of ABTS oxidation was monitored by measuring absorbance at 405 nm using BioTek Epoch microplate reader (Winooski, VT, USA). The linear portion of the absorbance (time-course) curves were used to calculate the slope, which corresponds to the rate (Δ absorbance/min) of the enzyme at each tested pH (as shown in S1 Fig). Each experiment was performed at least in triplicates.

For determination of optimum reaction temperature for the DyPs, experiments, pH 4 buffer, 70 μg DyP4, 1 mM ABTS, and 0.25 mM $H_2O_2$ were used. The reaction was carried out at different temperatures (20, 30 and 40°C) for 5 minutes, with the absorbance measured at 405 nm. Each experiment was performed in triplicates.

## DyP4 mediated degradation of EPs and transformation products identification

The extracted peak for the MRM transition of each EP was used to quantify the pollutants remaining after the enzymatic degradation by measuring the area under the curve (AUC) for

control and treated samples according to the following equation:

$$\% \text{ Percentage Remaining} : \frac{AUC}{AUC\,(control)}\;x\;100$$

AUC: treated sample containing DyP4, $H_2O_2$, buffer and EP (± HOBT) and AUC (control): DyP4, buffer and EP (± HOBT).

The transformation products after MBT degradation by the three DyPs were analyzed by LCMSMS, as previously described [32]. For that, pure samples of MBT [100 ppm] were analyzed by LCMSMS with and without enzymatic treatments and the potential transformation products (upon enzymatic treatment) were identified.

## Results and discussion

### LCMSMS method development

Multiple reaction monitoring (MRM) is a technique used in mass spectrometry to quantitate the amounts of specific molecules of interest, in this case, different EPs. Each EP is analyzed based on its mass using a quadrupole MS (Q1), which will undergo fragmentation in the collision cell, to generate product ions that are exclusive to the precursor, that will be monitored by a third quadrupole (Q3). The mass to charge (*m/z*) ratio that is observed for each EP and its corresponding product ion *m/z* ratio is referred to as an MRM transition. This technique allows specific detection of precursor to product transitions for multiple compounds simultaneously for quantitative analysis [33].

Multiple trials of manual tuning, directly to the MS, were carried out for each EP to identify the most suitable MRM transition. After obtaining the best MRM transition with its suitable CE and DP (Table 1), a method using LCMSMS was developed to run all the EPs at once while maintaining the detection specificity of each EP. All 15 EPs were run according to the method described in the material and methods section for 20 minutes. Fig 1 provides the total ion chromatogram at the MRM mode for the mix of the 15 EPs with insets showing paracetamol (152>110 m/z), caffeine (195>138 m/z) and 4-chloro-2-methylphenoxy acetic acid (199>141 m/z). Using this optimised method, all DyP4-mediated degradation experiments were carried out.

### pH and temperature optimization

It is well known that different environmental factors affect the rate of enzymatic reactions. Among these factors, pH and temperature are generally well understood. The majority of enzymes have an optimum pH at which the rate of the reaction will be at its maximum. The pH affects the enzyme by changing the ionization of the amino acids in the enzyme binding/active sites (as well as functional groups on the substrates), allowing for optimum binding and catalysis. However, if the enzyme is placed in an extreme acidic or basic condition, this may lead to denaturation and subsequent loss of enzymatic activity. Temperature on the other hand has a complex role on the enzyme's activity, as it can have a dual role affecting the enzyme. Raising the temperature increases the rate of the reaction, but at the same time, this will progressively lead to the inactivation of the enzyme caused by thermal denaturation [34].

For that reason, the optimum pH and temperature for WT, 3F6 and 4D4 were assessed. The results are summarized in Fig 2, which shows that WT activity peaks at pH 3 and at 30˚C, while the optimum pH and temperature for 3F6 and 4D4 were 4 and 20˚C, respectively, though both 3F6 and 4D4 maintained more than 75% activity at 30˚C as well. Our results are in agreement with previously published studies for wild type DyP4, which they showed that the optimum pH was 3.5 and the optimum temperature was 35˚C [27, 35]. These conditions were applied when performing subsequent degradation experiments.

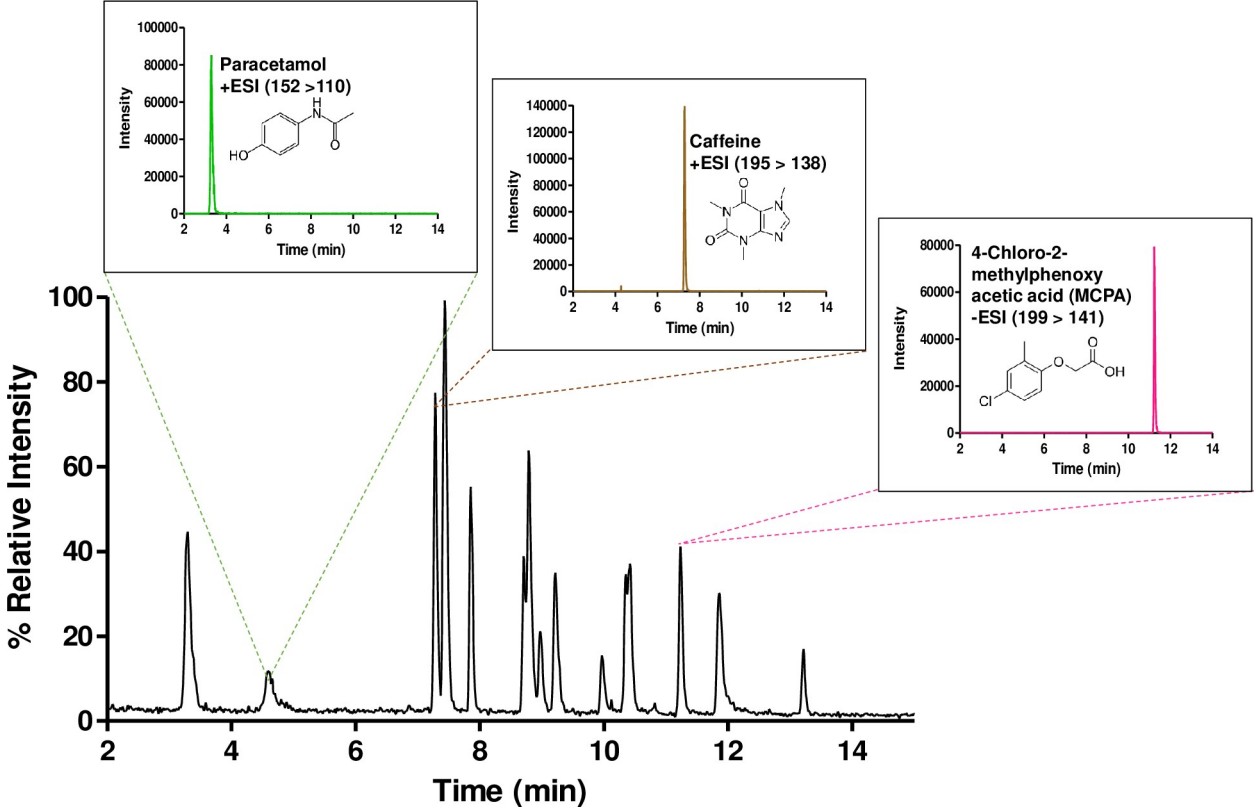

**Fig 1. LCMSMS MRM chromatogram for the combined 15 EPs at 0.25 ppm of each of them.** In addition to representative 3 EPs with their specific MRMs that are extracted from the main figure.

## DyP4 mediated degradation of EPs

Considering the optimum conditions for the enzymes' activities, degradation experiments were carried out to test the potency of WT, 3F6 and 4D4 in degrading various types of EPs.

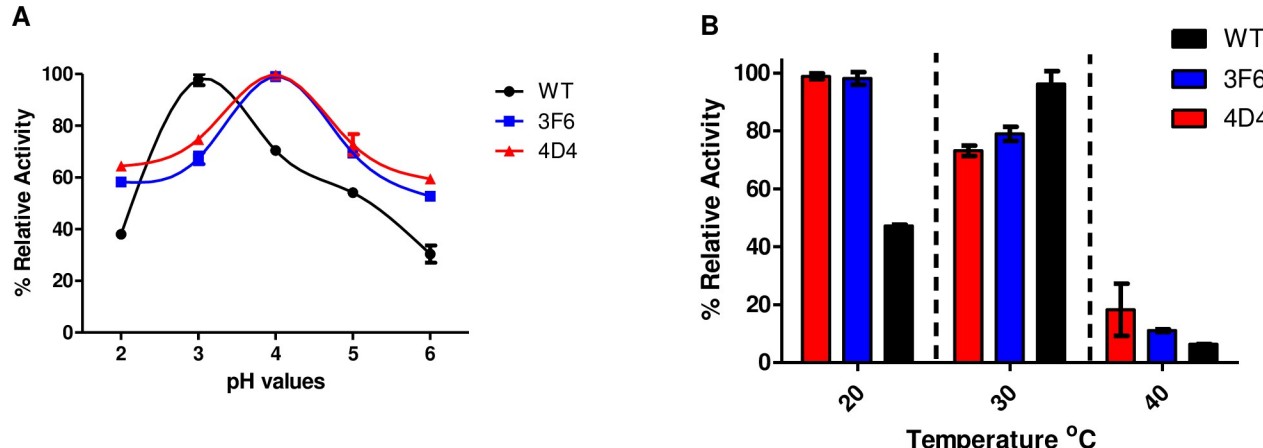

**Fig 2.** Relative ABTS oxidation activity of WT (black), 3F6 (blue) and 4D4 (red) at (A) pH (2–6) and (B) temperature (20-40°C) with ABTS concentration kept at 1mM. WT, 3F6 and 4D4 activities were normalized to the highest activity value (which was considered 100%) for each individual DyP4. Deviation values are standard deviations based on triplicate determinations.

**Table 2. Percent EP remaining after treatment with WT, 3F6 and 4D4, as described in material and methods.**

| Emerging Pollutant | WT | 3F6 | 4D4 | WT | 3F6 | 4D4 |
|---|---|---|---|---|---|---|
| | -HOBT | -HOBT | -HOBT | +HOBT | +HOBT | +HOBT |
| 2-Mercapto-benzothiazole | 0.6 ± 0.7 | 1.5 ± 0.9 | 2.8 ± 1.8 | 0.2 ± 0.5 | 0.0 ± 0.0 | 0.0 ± 0.0 |
| Paracetamol | 0.2 ± 0.1 | 3.5 ± 3.9 | 3.9 ± 5.5 | 0.3 ± 0.4 | 0.2 ± 0.4 | 0.3 ± 0.4 |
| Furosemide | 38.2 ± 4.2 | 36.4 ± 5.9 | 33.4 ± 4.2 | 0.0 ± 0.0 | 0.3 ± 0.2 | 0.0 ± 0.1 |
| Methylparaben | 103.5 ± 8.1 | 100.8 ± 4.0 | 103.4 ± 7.1 | 0.7 ± 0.5 | 0.5 ± 0.4 | 1.0 ± 1.2 |
| Sulfamethoxazole | 101.2 ± 4.2 | 95.3 ± 4.9 | 95.7 ± 6.6 | 2.1 ± 1.0 | 2.0 ± 0.3 | 2.1 ± 0.4 |
| Salicylic acid | 98.5 ± 8.6 | 99.0 ± 3.8 | 98.2 ± 4.2 | 0.4 ± 0.1 | 0.3 ± 0.3 | 0.5 ± 0.2 |
| Penicillin | 101.0 ± 6.0 | 94.8 ± 8.6 | 95.9 ± 5.4 | 91.3 ± 6.3 | 93.1 ± 6.7 | 95.4 ± 7.2 |
| Phenytoin | 117.8 ± 3.5 | 101.0 ± 3.8 | 106.8 ± 6.3 | 107.2 ± 4.6 | 108.6 ± 5.2 | 116.0 ± 6.6 |
| Trimethoprim | 101.1 ± 6.8 | 98.8 ± 5.2 | 98.7 ± 4.2 | 92.1 ± 4.6 | 90.6 ± 5.2 | 92.3 ± 3.3 |
| Chloramphenicol | 99.4 ± 6.9 | 97.9 ± 6.2 | 99.2 ± 4.0 | 97.3 ± 4.0 | 96.4 ± 2.2 | 99.0 ± 2.8 |
| Ibuprofen | 99.2 ± 6.3 | 91.0 ± 11.5 | 92.1 ± 8.7 | 93.1 ± 10.0 | 113.9 ± 2.4 | 94.5 ± 11.3 |
| Caffeine | 99.6 ± 6.0 | 99.3 ± 3.9 | 99.5 ± 4.2 | 99.0 ± 3.7 | 101.3 ± 5.6 | 98.6 ± 4.3 |
| Hydrochlorothiazide | 106.3 ± 10.3 | 99.0 ± 5.9 | 97.2 ± 6.2 | 89.1 ± 7.7 | 97.6 ± 10.8 | 94.5 ± 7.6 |
| 2-methyl-4-chlorophenoxyacetic acid | 98.6 ± 6.5 | 99.3 ± 5.2 | 96.3 ± 3.0 | 97.0 ± 3.1 | 100.5 ± 3.6 | 97.4 ± 3.5 |
| Ampicillin | 98.4 ± 8.2 | 97.2 ± 6.3 | 98.1 ± 6.3 | 96.6 ± 5.8 | 98.0 ± 5.1 | 95.4 ± 4.8 |

Averages of % EPs remaining after enzymatic treatment ± standard deviation are shown. Rows shaded in yellow show the degradation without HOBT, those in green show the degradation after the addition of HOBT, while the grey shaded rows represent the EPs that did not show any sign of degradation by DyP4.

For that, the mixture of 15 EPs were treated with DyP4 enzymes (WT, 3F6 and 4D4) in the presence or absence of $H_2O_2$ for 1 hour before being analyzed by LCMSMS. Negative control experiments using un-induced and induced cell lysates, in the absence of $H_2O_2$ and in the absence of HOBT were also conducted. Interestingly, the three enzymes were able to completely degrade 2-mercaptobenzothiazole (MBT) and paracetamol, while showing approximately 65% degradation of furosemide (Table 2 and Fig 3A). A recent study has reported that DyP4 was able to oxidize a range of substrates like ABTS, guaiacol and 2,6-dimethoxyphenol (DMP) in addition to its ability to decolorize a wide range of synthetic dyes such as anthraquinone, azo, and phenazine dyes [35]. However, to our knowledge, this is the first time DyP4 is shown to degrade emerging pollutants which highlights the potential use of DyP4 in industrial and environmental applications.

Peroxidases are known to oxidize a wide range of organic substrates in the presence of $H_2O_2$. However, sometimes, they require redox mediators to facilitate the oxidation-reduction reaction. Redox mediators are diffusible low molecular weight compounds that speed up peroxidase-based redox reactions by shuttling electrons between organic compounds (substrates) and the Compound I form of the peroxidase. Indeed, they have been shown to enhance the range of substrates that can be recognized by peroxidases and the increase the efficiency of degradation of recalcitrant compounds by several folds [36]. As shown in Fig 3A and Table 2, MBT and paracetamol were completely degraded, with or without 1-hydroxybenzotriazole (HOBT), a commonly used redox mediator. Interestingly, furosemide, which showed about 65% degradation without HOBT, appeared to be completely degraded when HOBT was included in the reaction mixture.

Even more dramatic and remarkable was the observation that, HOBT also expanded the substrate repertoire of DyP4. As can be seen in Table 2, salicylic acid, methyl paraben and sulfamethoxazole, which were completely recalcitrant to degradation by DyP4-WT, or the evolved variants, could be efficiently degraded by all three DyP4s in the presence of HOBT.

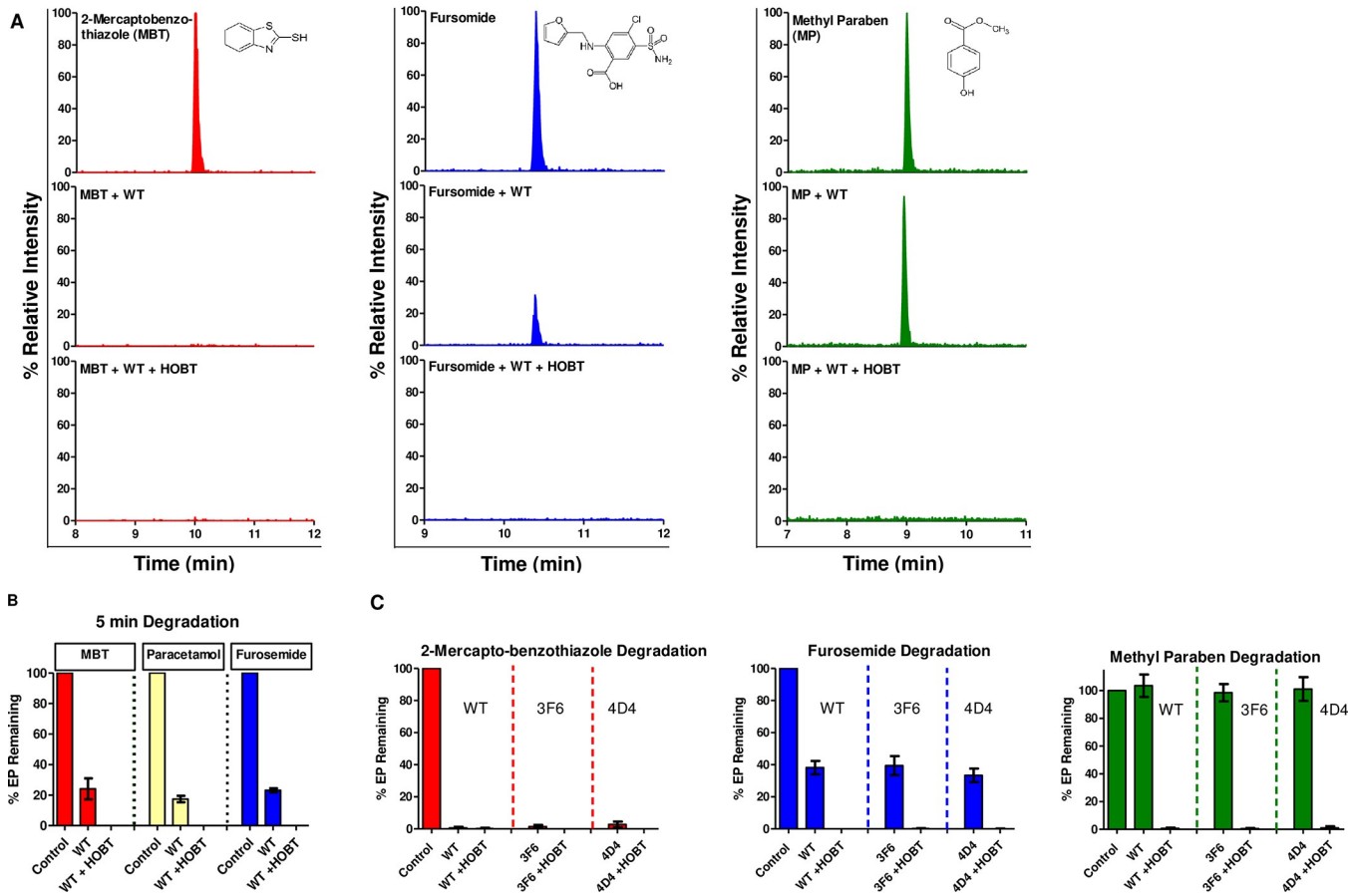

**Fig 3.** (A) Individual MRM chromatograms of three emerging pollutants (MBT, furosemide and methylparaben) upon DyP4 treatment. First panel represents the EP control, the second panel represents the % relative intensity for each EP after the addition of WT and last panel represents the % relative intensity for each EP after the addition of WT with HOBT simultaneously. (B) Percentage of three EPs (MBT, paracetamol and furosemide) degradation upon treatment with WT after 5 minutes. (C) Percentage of three EPs (MBT, furosemide and methylparaben) degradation upon treatment with WT, 3F6 and 4D4 only or with HOBT.

This is consistent with our earlier studies that showed the synergistic effects of HOBT in enhancing EP degradation by soybean peroxidase and chloroperoxidase [32, 37].

We also wanted to test if the rate of EP degradation will be enhanced by HOBT especially for the three completely degraded EPs (Table 2). Indeed, HOBT addition enhanced the DyP4-WT degradation efficiency (from ~ 80% degradation to 100% degradation) and reduced the time needed for a complete degradation of MBT, paracetamol and furosemide to just 5 minutes (Fig 3B). In comparison, the same reaction without HOBT required longer incubation time for the EPs to be degraded—MBT needed about 30 minutes while paracetamol required 60 minutes for full degradation. As for furosemide, only 62% degradation was observed after 60 minutes. This interesting result underscores the potential synergistic effects that redox mediators can confer in peroxidase-mediated reactions.

Fig 3C summarizes the type of effects seen in our HOBT experiments–the redox mediator could enhance the degradation of 1) EP that were completely degraded by DyPs alone (made them degrade faster–Fig 3B), 2) EPs that were mostly degraded (~ 65%) by DyPs alone, or 3) EPs that were recalcitrant to degradation by DyPs alone. It should be mentioned that HOBT is known to have aquatic toxicity and hence will not be a suitable redox mediator to use in real-

**Table 3. Summary of previous MBT degradation studies, showing the reported transformation products as well as those found in the current study.**

| Remediation Approach | Main findings (transformation products) | Transformation products (m/z) | Reference (Chronological order) |
|---|---|---|---|
| Ozonation | Several products including Benzothiazole (BT), 2(3H)-Benzothiazolone (OBT), benzothiazole-2-sulfite, hydroxybenzothiazole (OHBT) | 69, 78, 95, 96, 106, 108, 109, 122, 123, 135, 136, 150, 151, 165, 214 | [42] |
| **Bacterial strain (*Rhodococcus rhodochrous* OBT18)** | ***cis*-dihydrodiol MBT, 6-hydroxy MBT, Diacid MBT** | **182, 200, 230** | **[43]** |
| UV irradiation | Nine detected products including BT, 2,2'-thiobisbenzothiazole, 2-mercaptobenzothiazole disulfide | 139, 150, 269, 301, 333 | [44] |
| UV irradiation catalyzed by sodium decatungstate (Na4W10O32) | Six products including one dimer, four isomers of OH-MBT, sulfoxide MBT derivative | 184, 200, 333 | [45] |
| UV-irradiation | Dimeric compounds in aqueous and organic solutions | 185, 228, 301, 348,333 | [46] |
| Imprinted photocatalyst based on Fe3O4/g-C3N4 | Several products including BT | 108, 125, 134, 139 | [47] |
| **Plant assimilation by *Arabidopsis thaliana*** | **Glycosylated MBT, three amino acid conjugated MBT compounds** | **254, 296, 329, 340** | **[48]** |
| **Soybean peroxidase and Chloroperoxidase** | **Five products by Soybean Peroxidase and two products by Chloroperoxidase** | **120, 136, 182, 301, 333** | **[37]** |
| **Bacterial strain (*Alcaligenes sp.* MH146 strain CSMB1)** | **Five products including 2-BT, 2-OHBT, 2-methylated MBT** | **54, 80, 95, 123, 136, 150, 151, 163, 176, 201** | **[49]** |
| Photodegradation by photocatalyst, g-C$_3$N$_4$ modified with Fe$_3$O$_4$ Quantum Dots | Six products including monocyclic compounds | 82, 110, 114, 125, 139, 171 | [50] |
| Photo-Fenton process using natural clay powder | Several products including OH-MBT, BT, OH-BT | 135, 149, 151, 152, 166, 167, 173, 183, 198, 199, 214, 215, 285, 296, 314, 330, 332 | [51] |
| bxiochar-based photocatalyst of g-C3N4-C, | Seven products including monocyclic compounds | 56, 80, 108, 122, 155, 171, 184 | [52] |
| 9-Bi$_2$WO$_6$ | Several products including monocyclic compounds | 82, 110, 114, 125, 139, 171 | [53] |
| Nano-cubes/ | | | |
| In (OH)$_3$ photocatalyst | | | |
| **Microbial electrolysis cells (MECs)** | **Twenty-five products including dimers and monomers** | **93, 94, 108, 109, 110, 125, 126, 135, 151, 158, 167, 169, 174, 183, 187, 199, 215, 231, 268, 283, 284, 300, 332, 364** | **[40]** |
| UV irradiation | Twenty-two by-products including isomers of OH-BT | 122,134, 136, 150, 152, 166, 168, 182, 204, 216, 230, 246 | [12] |
| Charcoal and graphitic carbon nitride-based photocatalyst | Several products including monocyclic compounds | 72, 84, 93, 127 | [54] |
| Modified carbon nitride photocatalyst | Six products including monocyclic compounds | 82, 110, 124, 136, 155 | [55] |
| **Fungal DyP4 peroxidases** | **Six products including dimeric and monomeric compounds** | **136, 166, 268, 284, 333** | **This study** |

Published data from biological approaches are highlighted in grey.

life wastewater treatment applications. There are other less toxic and better alternatives available, which would be better suited to field applications.

## LCMSMS analysis of MBT transformation products upon DyP4 treatment

The nature, identity, and mechanisms by which the transformation products are generated in these enzymes mediated EP degradation studies are other key questions that are being explored by the scientific community. As would be expected that EP-derived transformation products can be structurally diverse compounds that could be formed through different

**Fig 4. Proposed transformation products of MBT after degradation with DyP4-WT, 3F6 and 4D4.**

conversion pathways, and hence their identification is crucial for health and environmental reasons [38].

For example, a recent study was conducted to evaluate the toxicity of seven compounds that belonged to different therapeutic groups along with the toxicity of their main transformation products on five organisms. The report concluded that the toxicity of the transformation products was similar or lower compared to their parent compounds. However, some transformation products showed higher toxicity in some organisms compared to their parent compounds [39]. Since less is currently known about the occurrence and the fate of transformation products, additional studies are required to investigate the transformation products of EPs, their occurrence, toxicity, and potential environmental risk.

In the current study, an attempt was carried out to identify the transformation products of MBT upon treatment with the three DyP4 enzymes. Additionally, the transformation products formed when MBT was degraded by DyP4 were compared to reported treatments in Table 3. All the MBT degradation products found in this study have been previously reported in literature and are shown in Fig 4 (with their potential structures). The transformation products observed in our current study are consistent with previous studies in which different biological methods (microbes and pure enzymes) were used to treat MBT. For example, the 166 m/z, 268 m/z and 284 m/z species reported here were also detected during MBT degradation in microbial electrolysis cells [40]. Additionally, the 136 m/z and 333 m/z transformation products have been previously reported during the degradation of MBT by pure soybean peroxidase as well as chloroperoxidase enzymes [37]. Table 3 shows a summary of previously published studies on the use of different mitigation approaches to degrade MBT and some of the transformation products identified after each treatment. It is interesting to note that all of the 5 transformation products identified in our current study have been previously reported in

literature with other degradation approaches, thus suggesting that perhaps bioremediation and catalytic oxidative reactions share similar mechanistic pathways.

Although our present work and previously published studies show efficient transformation and degradation of MBT into various species, one has to be mindful about potential residual toxicities of these intermediate compounds. For example, a recent study that has identified the transformation products (one is similar to our study) generated during UV-treatment of MBT, showed significant aquatic toxicity and potential hazard to human health [12]. This underscores the importance of such toxicity studies to understand and assess the potential risk of transformation products in order to design and develop relevant mitigation strategies. In cases where bioremediation generates transformation products with residual toxicities, additional chemical and/or physical approaches need to be used in tandem to completely degrade and detoxify organic pollutants.

## Conclusion

The present study investigated the potential use of DyP4 and its evolved variants as remediating agents to degrade a panel of 15 EPs using a sensitive and robust LCMSMS method. Our results show that MBT, paracetamol and furosemide were efficiently degraded by DyP4 and $H_2O_2$ alone. When HOBT redox mediator was added, the time required for a complete degradation of these three EPs could be shortened to just 5 minutes. HOBT also enabled DyP4 (and evolved variants) to degrade three additional EPs (methyl paraben, sulfamethoxazole and salicylic acid). Further investigation was carried for MBT degradation which elucidated five transformation products generated during their degradation that were reported previously. Additionally, we showed that evolved variants of DyP4 (with significantly higher $H_2O_2$-tolerance) were equally effective in degrading various EPs as the wild type peroxidase. This finding is very significant as one of the major limitations of using peroxidases for bioremediation is their potential oxidation and denaturation by the $H_2O_2$ that is needed for their activity. Results presented here suggest that enzyme engineering approach that was used to generate $H_2O_2$-tolerant variants of DyP4 peroxidases (3F6 and 4D4) can generate more robust, stable, and powerful bioremediation agents. The obvious next step would be to scale-up enzyme-mediated remediation approaches by immobilizing cheaply produced recombinant enzymes (or their improved variants) onto solid supports to create bioreactors/columns and test them on real-life wastewater. We have recently shown that soybean peroxidase can be efficiently supported on photocatalytic supports to produce novel hybrid biocatalysts more potent than the enzyme or the photocatalysts [41]. The current study describes the first instance of the use of recombinant DyP4 fungal peroxidases (and its evolved variants) for bioremediation of organic pollutants and opens the door for the use of engineered peroxidases for such applications.

## Supporting information

**S1 Fig.** A) Oxidation of ABTS at pH 6 by DyP4 wild-type (in triplicates). The inset shows the linear regression of the linear portion of the curve, with the appropriate equation and correlation data. B) Oxidation of ABTS by DyP4 (wild-type) at pH values from 2–9 (average of triplicates shown).
(TIF)

## Author Contributions

**Conceptualization:** Syed Salman Ashraf.

**Data curation:** Khawlah Athamneh.

**Funding acquisition:** Syed Salman Ashraf.

**Investigation:** Khawlah Athamneh, Aysha Alneyadi, Aya Alsadik.

**Project administration:** Syed Salman Ashraf.

**Resources:** Tuck Seng Wong.

**Writing – original draft:** Khawlah Athamneh.

**Writing – review & editing:** Tuck Seng Wong, Syed Salman Ashraf.

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
