## [Decision Letter · Decision Letter 0]

2 Dec 2021

PONE-D-21-27995Efficient degradation of various emerging pollutants by wild type and evolved fungal DyP4 peroxidasesPLOS ONE

Dear Dr. ashraf,

Thank you for submitting your manuscript to PLOS ONE. After careful consideration, we feel that it has merit but does not fully meet PLOS ONE’s publication criteria as it currently stands. Therefore, we invite you to submit a revised version of the manuscript that addresses the points raised during the review process.

We look forward to receiving your revised manuscript.

Kind regards,

Muhammad Adrees, Ph.D.

Academic Editor

PLOS ONE

Journal Requirements:

Reviewers' comments:

Reviewer's Responses to Questions

**Comments to the Author**

1. Is the manuscript technically sound, and do the data support the conclusions?

Reviewer #1: Yes

Reviewer #2: Yes

2. Has the statistical analysis been performed appropriately and rigorously? 

Reviewer #1: No

Reviewer #2: Yes

3. Have the authors made all data underlying the findings in their manuscript fully available?

Reviewer #1: No

Reviewer #2: Yes

4. Is the manuscript presented in an intelligible fashion and written in standard English?

Reviewer #1: Yes

Reviewer #2: Yes

5. Review Comments to the Author

Reviewer #1: I have gone through the article"Efficient degradation of various emerging pollutants by wild type and evolved fungal DyP4 peroxidases" This is an interesting topic that is closely related to the scope of PLOS ONE. However, the manuscript should be improved in a number of following ares:

1- The intorduction needs to be enhanced. More recent references are needed and some some reference which is not related to this article should be avoided.

2- It is recommended to discuss and compare the current study results with previously published studies in a TABLE form.

3-It is suggested to write the equations and provide the graph of linear portion of absorbance Curve to calculate the Slope as the representation on line 146-147(May be as Supplementary Information).

Reviewer #2: I have gone through the article "Efficient degradation of various emerging pollutants by wild type and evolved fungal DyP4 peroxidases" This is an Interesting Topic that is closely related to the scope of PLOS ONE. However, minor revisions are suggested for following areas:

1-Intriduction need to be improved by removing irrelevant information, old references and by the addition of some recent studies.

2- compare the results of your study with previous studies in a separate table (year of study, researcher, methodology, main findings)

3-Please write the equations in equation form for a clear understanding

4-Conclusion is need to be revised

5- Define how this study will be beneficial on commercial scale

6. PLOS authors have the option to publish the peer review history of their article (what does this mean?). If published, this will include your full peer review and any attached files.

Reviewer #1: No

Reviewer #2: **Yes: **Dr. Fariha Jabeen

---

## [Author Response · Author response to Decision Letter 0]

16 Dec 2021

PONE-D-21-27995

Efficient degradation of various emerging pollutants by wild type and evolved fungal DyP4 peroxidases

Response to Reviewers

Reviewer #1: I have gone through the article "Efficient degradation of various emerging pollutants by wild type and evolved fungal DyP4 peroxidases" This is an interesting topic that is closely related to the scope of PLOS ONE. However, the manuscript should be improved in a number of following areas:

1- The introduction needs to be enhanced. More recent references are needed and some reference which is not related to this article should be avoided.

Response: Thank you for your comment. As requested, new recent references have been added (lines 46, 52, 67, 79, 82, 90). Some of the old references have also been avoided (lines 42, 46, 53). Unrelated references have also been removed.

2- It is recommended to discuss and compare the current study results with previously published studies in a TABLE form. 

Response: Thank you very much for pointing this out. A table comparing previously published findings on 2-mercaptobenzothiazole degradation to the results of this study has been made and added to the manuscript.

3-It is suggested to write the equations and provide the graph of linear portion of absorbance Curve to calculate the Slope as the representation on line 146-147(May be as Supplementary Information).

Response: Thank you for your comment. We have included a supplemental figure that shows a representative data analysis, including linear regression and data fitting with the appropriate equations for determining the slope. 

 

Reviewer #2: I have gone through the article "Efficient degradation of various emerging pollutants by wild type and evolved fungal DyP4 peroxidases" This is an Interesting Topic that is closely related to the scope of PLOS ONE. However, minor revisions are suggested for following areas:

1-Introduction need to be improved by removing irrelevant information, old references and by the addition of some recent studies.

Response: Thank you very much for commenting on this. New recent references have been added. Some of the old references have also been avoided. Unrelated references have also been removed.

2- compare the results of your study with previous studies in a separate table (year of study, researcher, methodology, main findings)

Response: Thank you for your comment. A table comparing previously published findings on 2-mercaptobenzothiazole degradation to the results of this study has been made and added to the manuscript.

3-Please write the equations in equation form for a clear understanding

Response: Thank you very much for your comments. The equation representing % percentage remaining has been written in an equation form. 

4-Conclusion is needed to be revised

5- Define how this study will be beneficial on commercial scale

Response: Thank you for this suggestion. As requested, the conclusion has been significantly expanded and now also includes discussion (pasted below) about the future application of this technology on commercial scale.

“This finding is very significant as one of the major limitations of using peroxidases for bioremediation is their potential oxidation and denaturation by the H2O2 that is needed for their activity. Results presented here suggest that enzyme engineering approach that was used to generate H2O2-tolerant variants of DyP4 peroxidases (3F6 and 4D4) can generate more robust, stable, and powerful bioremediation agents. The obvious next step would be to scale-up enzyme-mediated remediation approaches by immobilizing cheaply produced recombinant enzymes (or their improved variants) onto solid supports to create bioreactors/columns and test them on real-life wastewater. We have recently shown that soybean peroxidase can be efficiently supported on photocatalytic supports to produce novel hybrid biocatalysts more potent than the enzyme or the photocatalysts ((Morsi et al., 2021).”

---

## [Decision Letter · Decision Letter 1]

27 Dec 2021

Efficient degradation of various emerging pollutants by wild type and evolved fungal DyP4 peroxidases

PONE-D-21-27995R1

Dear Dr. ashraf,

We’re pleased to inform you that your manuscript has been judged scientifically suitable for publication and will be formally accepted for publication once it meets all outstanding technical requirements.

Kind regards,

Muhammad Adrees, Ph.D.

Academic Editor

PLOS ONE

Additional Editor Comments (optional):

Reviewers' comments:

Reviewer's Responses to Questions

**Comments to the Author**

1. If the authors have adequately addressed your comments raised in a previous round of review and you feel that this manuscript is now acceptable for publication, you may indicate that here to bypass the “Comments to the Author” section, enter your conflict of interest statement in the “Confidential to Editor” section, and submit your "Accept" recommendation.

Reviewer #1: All comments have been addressed

Reviewer #2: All comments have been addressed

2. Is the manuscript technically sound, and do the data support the conclusions?

Reviewer #1: Yes

Reviewer #2: Yes

3. Has the statistical analysis been performed appropriately and rigorously? 

Reviewer #1: Yes

Reviewer #2: Yes

4. Have the authors made all data underlying the findings in their manuscript fully available?

Reviewer #1: Yes

Reviewer #2: Yes

5. Is the manuscript presented in an intelligible fashion and written in standard English?

Reviewer #1: Yes

Reviewer #2: Yes

6. Review Comments to the Author

Reviewer #1: Authors addressed all the comments and revised the manuscript accordingly. I recommend the acceptance of manuscript.

Reviewer #2: (No Response)

7. PLOS authors have the option to publish the peer review history of their article (what does this mean?). If published, this will include your full peer review and any attached files.

Reviewer #1: No

Reviewer #2: **Yes: **Dr. Fariha Jabeen

---

## [Editor Report · Acceptance letter]

4 Jan 2022

PONE-D-21-27995R1 

Efficient degradation of various emerging pollutants by wild type and evolved fungal DyP4 peroxidases 

Dear Dr. Ashraf:

I'm pleased to inform you that your manuscript has been deemed suitable for publication in PLOS ONE. Congratulations! Your manuscript is now with our production department. 

Kind regards, 

on behalf of

Dr. Muhammad Adrees 

Academic Editor

PLOS ONE